# Rank1: Test-Time Compute for Reranking in Information Retrieval

**Orion Weller**    **Kathryn Ricci**    **Eugene Yang**

**Andrew Yates**    **Dawn Lawrie**    **Benjamin Van Durme**

Johns Hopkins University

oweller@cs.jhu.edu

## Abstract

We introduce RANK1, the first reranking model trained to take advantage of test-time compute. RANK1 demonstrates the applicability within retrieval of using a reasoning language model (i.e. OpenAI's o1, Deepseek's R1, etc.) for distillation in order to rapidly improve the performance of a smaller model. We gather and open-source a dataset of more than 600,000 examples of R1 reasoning traces from queries and passages in MS MARCO. Models trained on this dataset show: (1) state-of-the-art performance on advanced reasoning and instruction following datasets; (2) work remarkably well out of distribution due to the ability to respond to user-input prompts; and (3) have explainable reasoning chains that can be given to users or retrieval-augmented generation (RAG) systems. Further, we demonstrate that quantized versions of these models retain strong performance while using less compute/memory. Overall, RANK1 shows that test-time compute allows for a fundamentally new type of explainable and performant reranker model for search.[1]

## 1 Introduction

Reasoning language models (LMs) like OpenAI's o1, Deepseek's R1, and Gemini's Flash-Thinking have shown improved reasoning abilities through the use of test-time compute, i.e. generating a *reasoning chain* of tokens that allow the model to "think" before giving the final answer. Another benefit to these style of models is that the reasoning chain can easily be distilled into smaller models. As shown by Deepseek (Guo et al., 2025) smaller models learn incredibly well from simple supervised fine-tuning on the larger model's reasoning chains.

The benefits that reasoning models bring to general text generation would also be valuable in an information retrieval (IR) context: allowing models additional time to reason why a passage could be relevant, while also allowing an auditable reasoning process to give to the user or RAG system. For this approach to be maximally effective, the model must be able to reason over both query and passage; if applied solely to the query, the reasoning model would not know the passage context and would have to try to infer it. This would be a form of query-expansion (Nogueira et al., 2019) and limits the model's ability to be precise. Thus, our work focuses on bringing test-time compute to IR in a *reranking* setting, where the model needs to compute the relevance of an initial top-k candidates.

To accomplish this goal, we sample 635,000 examples of R1's thought process on the MS MARCO dataset (Nguyen et al., 2016). We then fine-tune a suite of LMs on these reasoning chains and find that they show remarkable reasoning capabilities. Surprisingly, they also exhibit an ability to be prompted despite training from the base LMs only (without instruction fine-tuning) and while having no instruction-based IR training data (only MS MARCO).

---

[1]Models, code, and data are available at https://github.com/orionw/rank1

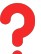 what county is colton in

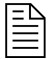 Colton, California. Colton is a city in San Bernardino County, California, United States...

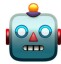 <think> The user is asking "what county is Colton in." ... So the first thing I notice is that the passage directly mentions San Bernardino County as the county where Colton is located.

The query is specifically about the county, and the answer is right there in the first sentence of the passage.

But wait, maybe I should check if there's any possibility that the passage isn't relevant. The rest of the passage talks ... but the key information about San Bernardino County is present ...

So the answer should be true because the passage answers the query directly.

</think> **true**

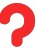 do snow leopards change color

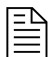 Snow leopards have long, thick fur, and their base color varies from smoky gray to yellowish tan, with whitish underparts ...

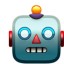 <think> The user is asking if snow leopards change color. The passage provided describes their fur color as varying from smoky gray to yellowish tan, with some patterns. Hmm, the key here is whether "varies" implies that they change color over time, like seasonally or due to some factors.

But wait, "varies" might just mean that different individuals have different base colors, not that a single leopard changes its color. The passage doesn't mention anything about changing color; it's just describing their appearance. ...

Therefore, the passage isn't directly relevant to the question of whether they change color. So the answer should be false.

</think> **false**

Figure 1: Example reasoning traces from R1, used to train RANK1. Note the self-inquisitory reasoning (in blue) where the model questions if it has the correct answer.

This includes state-of-the-art performance on the BRIGHT benchmark for reasoning (Su et al., 2024), the NevIR benchmark on complex negation understanding, and the mFol-lowIR dataset on multilingual instruction-following in IR – despite having no non-English reasoning training data (Weller et al., 2025a).

We also conduct a detailed analysis of performance on traditional IR benchmarks, such as TREC DL19 (Craswell et al., 2020) and BEIR (Thakur et al., 2021). We find that these datasets are likely model-saturated, as RANK1 surfaces an extremely large number of unjudged documents (364% more than RankLLaMA-14B). We argue these benchmarks are no longer helpful for distinguishing between the best performing rerankers and that focus should be put on benchmarks that examine advanced reasoning, instruction-following, and have more modern (e.g. post-ChatGPT) annotations.

Overall, RANK1 shows the benefits that test-time compute can bring to the field of IR: **explainable reasoning chains** that can be audited by users or used by agentic RAG systems, **significantly improved reasoning** performance, and **adaptability from user-given prompts**.

## 2 Model Training

### 2.1 Data Preparation

In order to distil from R1, we first need to gather data to use for prompting it. We use the MS MARCO collection (Nguyen et al., 2016) due to its diversity in topics and common use in previous work. We use kluster.ai as the API service to access R1 using their batch mode with a temperature of 0.3 and a maximum of 1000 output tokens.[2]

We generate data from an equal number (25% of the data) from each of the (1) positive examples in MS MARCO, (2) sampled negatives from Tevatron[3] (gathered from BM25 and CoCondenser), (3) rank 1-5 hard negatives from mT5-13B, and (4) rank 5-10 hard negatives from mT5-13B. However, we found that R1 classified roughly 80% of the mT5 hard negatives as positives. Thus we did another round of generation using only hard negatives from rank

---

[2]In our initial testing we found that all reasoning chains were shorter than 1000 tokens. As each MS MARCO passage is typically around 100-200 token, R1 doesn't need more than 1000 tokens to reason over it.

[3]From https://huggingface.co/datasets/Tevatron/msmarco-passage-aug

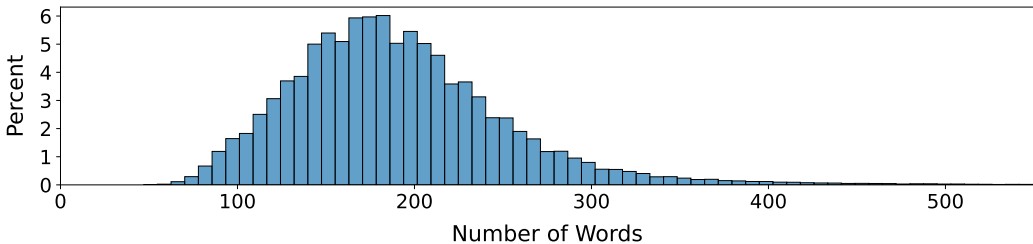

Figure 2: Distribution of word lengths of the reasoning chains generated from R1. It has a slightly rightward skew but is generally normal shaped. Note that there is no noticeable difference in the distribution between passages that are predicted relevant vs non-relevant.

5-10 and easy negatives from Tevatron. As the mined hard negatives from mT5 do not have an official label, it is likely that many of them were false negatives and that R1 was correct.

After all generation was done, our dataset has 635,264 examples of R1 generations, where R1 labeled 62.9% as relevant and 37.1% as non-relevant. We show a plot of these generation lengths in Figure 2 where we see a fairly normal distribution. Although we thought there may be length differences between these four subsets of data, we found that they all had the same rough distribution.

## 2.2 Data Mix and Quality Filtering

Since we had a surplus of documents judged relevant, we tried various methods to arrive at our final data mix. We initially tried using all the data, after balancing for the labels. We found that this performed significantly worse than filtering based on the labels we were most sure about (i.e. the positives from MS MARCO and the negatives from Tevatron). However, even on those subsets, 15% of R1's final prediction disagreed with the implied labels – thus we filtered out these instances.

Beyond being labeled as positives, we found that a large number of the mT5 mined hard negative samples were noisy.[4] To alleviate this, we used a model trained on the first mix to self-filter the data.[5] This filtered another 10% of the data, mostly false negatives. Since we still had a surplus of "relevant" labeled instances, we took all positives from the official MS MARCO positives[6] and all negatives from the self-filtered set. This left us with a training set of 386,336 high quality training samples: 136k from the original MS MARCO positives, 154k from the Tevatron negatives, and 96k from the mT5 negatives. We note that having more negatives than positives is standard: RankLLaMA (Ma et al., 2024) trained on a 15:1 ratio of negatives to positives for 7 million examples.

## 2.3 Training

We train three main models from the Qwen 2.5 family of models (Yang et al., 2024). These models have shown improved performance on recent LM benchmarks and have a wide range of models, allowing us to show the effects of scale. We use the 7B, 14B, and 32B parameter models.[7] We show that alternative base models are also effective in Section 4. During initial experiments we found that the base models outperformed their instruction-tuned variants, so all models are trained from the base version (e.g. no instruction-training).[8]

---

[4]See Appendix Section B where we found that 2/3rds of the mT5 negatives were actually positives.

[5]We used the RANK1 Mistral 24B version as it was the largest that fit on 1 GPU, filtering all instances where the model's prediction didn't agree with R1's prediction.

[6]From the MS MARCO small triples file.

[7]We show ablations on smaller Qwen 2.5 models in Section 3.8.

[8]This could be because instruct-versions are optimized for chat and math data, whereas we have a large amount of reranking-specific data that doesn't benefit from chat/math-based instructions.

| | StackExchange | | | | | | | Coding | | Theorem-based | | | Avg. |
|---|---|---|---|---|---|---|---|---|---|---|---|---|---|
| | Bio. | Earth. | Econ. | Psy. | Rob. | Stack. | Sus. | Leet. | Pony | AoPS | TheoQ. | TheoT. | |
| BM25 | 19.2 | 27.1 | 14.9 | 12.5 | 13.5 | 16.5 | 15.2 | 24.4 | 7.9 | 6.0 | 13.0 | 6.9 | 14.8 |
| BM25 on GPT-4o CoT | 53.6 | 53.6 | 24.3 | 38.6 | 18.8 | 22.7 | 25.9 | 19.3 | 17.7 | 3.9 | 18.9 | 20.2 | 26.5 |
| MonoT5-3B | 16.0 | 24.0 | 17.7 | 19.5 | 8.0 | 10.5 | 19.5 | 17.2 | 29.2 | 7.1 | 20.3 | 12.0 | 16.8 |
| RankLLaMA-7B | 17.5 | 15.5 | 13.1 | 13.6 | 17.9 | 6.9 | 16.9 | 8.4 | **46.8** | 2.2 | 4.5 | 3.5 | 13.9 |
| RankLLaMA-13B | 21.6 | 19.1 | 16.3 | 14.0 | 15.7 | 7.7 | 18.5 | 8.8 | 31.1 | 1.7 | 4.4 | 4.9 | 13.7 |
| Rank1-7B | 48.8 | 36.7 | 20.8 | 35.0 | 22.0 | 18.7 | **36.2** | 12.7 | 31.2 | 6.3 | **23.7** | 37.8 | 27.5 |
| Rank1-14B | 49.3 | **37.7** | **22.6** | 35.2 | **22.5** | 20.8 | 33.6 | 17.7 | 33.2 | 8.4 | 22.5 | 41.4 | 28.7 |
| Rank1-32B | **49.7** | 35.8 | 22.0 | **37.5** | **22.5** | **21.7** | 35.0 | **18.8** | 32.5 | **10.8** | 22.9 | **43.7** | **29.4** |

Table 1: The performance of retrieval models on BRIGHT. BM25 scores are taken from the official BRIGHT paper. All models rerank from the BM25 on GPT-4o CoT top 100 documents, but are not given the GPT-4o CoT. **We find a large gap between similar sized rerankers and RANK1 models (sometimes 2x)**. Bold indicates the best score in the reranker section.

We train the models with LoRA using LLaMA-Factory (Zheng et al., 2024) for up to two epochs or for up to three days. We found that there was increased learning for roughly the 1.5 epochs but then performance saturated. For more details and hyperparameters configurations see Appendix A.

## 3 Experiments

We show the capabilities of the RANK1 through evaluation on advanced reasoning, instruction-following, and semantic-understanding datasets. We also demonstrate performance on traditional benchmarks. We use `mteb` (Muennighoff et al., 2022; Enevoldsen et al., 2025) to run all experiments except for DL19, which uses `rankllm` (Pradeep et al., 2023a). Inference is powered by `vllm` (Kwon et al., 2023) which makes it significantly faster than vanilla `transformers` (Wolf et al., 2020).

### 3.1 Baselines

We use BM25 (Robertson et al., 1994; Lù, 2024) and mE5-base (Wang et al., 2024) for our first stage models. For reranking models we focus on other pointwise reranking models (e.g. models that output a score per document): MonoT5-3B (Nogueira et al., 2019), mT5-13B fine-tuned on MMARCO for multilingual tasks (Jeronymo et al., 2023), and RankLLaMA 7 and 13B (Ma et al., 2024). For instruction following tasks we also include FollowIR-7B (Weller et al., 2024a) which was trained solely for instruction-following.

We show results for listwise models when those scores are available (Pradeep et al., 2023b; Sun et al., 2023), but note that they are not comparable – listwise models take an order of magnitude more time at inference due to their sequential dependencies and have the advantage of seeing all documents in their context when reranking. We show them generally as a strong upper bound, as when state-of-the-art (SOTA) LMs (i.e. GPT-4o) are used.

### 3.2 Reasoning Capabilities

We show results on the reasoning intensive BRIGHT benchmark (Su et al., 2024) in Table 1. All reranker models judge the top 100 documents found using BM25 on the query plus GPT-4o's Chain of Thought (CoT) reasoning, which performed significantly better than BM25 without the query expansion (thus including more relevant documents in the top 100 to test rerankers with). However, at inference time, the rerankers are not given the CoT. We see a large gap between RANK1 and other models: in many cases near double the nDCG@10 score (e.g. 18.7 vs 7.7 for RANK1-7B vs RankLLaMA-13B on Stackoverflow).

These results are especially notable when you consider that RANK1 models were **trained on an order of magnitude less data** than models like RankLLaMA (7 million vs 600k) **while using the same training dataset** (MS MARCO). We also find that performance scales with model size, with the 32B model outperforming the smaller models. Thus, RANK1 is SOTA when reasoning is needed.

| Model | Persian | | Chinese | | Russian | | Average | |
|---|---|---|---|---|---|---|---|---|
| | nDCG@20 | p-MRR | nDCG@20 | p-MRR | nDCG@20 | p-MRR | nDCG@20 | p-MRR |
| mE5-base | 0.289 | -3.9 | 0.316 | +3.4 | 0.307 | -2.1 | 0.304 | -0.9 |
| MonoT5-3B | 0.118 | -2.4 | 0.231 | +5.0 | 0.240 | +6.8 | 0.196 | +3.1 |
| FollowIR-7B | 0.225 | +1.8 | 0.375 | +8.7 | 0.376 | +0.4 | 0.325 | +3.7 |
| mT5-13B | 0.453 | -0.7 | 0.474 | +2.3 | 0.505 | -0.6 | 0.477 | +0.4 |
| RankLLaMA 7B | 0.229 | +0.8 | 0.272 | +1.1 | 0.248 | +0.1 | 0.250 | +0.7 |
| RankLLaMA 13B | 0.256 | +0.8 | 0.287 | +1.8 | 0.320 | -0.6 | 0.288 | +0.6 |
| Rank1-7B | 0.564 | +7.0 | 0.582 | +3.1 | 0.511 | -0.0 | 0.552 | +3.4 |
| Rank1-14B | **0.572** | **+11.9** | **0.611** | +4.6 | 0.516 | +5.4 | **0.567** | **+7.3** |
| Rank1-32B | 0.555 | +3.9 | 0.598 | **+4.9** | **0.521** | **+6.6** | 0.558 | 5.2 |

Table 3: mFollowIR Cross-Lingual scores across three language subsets. Bold indicates best score.

| Model | Persian | | Chinese | | Russian | | Average | |
|---|---|---|---|---|---|---|---|---|
| | nDCG@20 | p-MRR | nDCG@20 | p-MRR | nDCG@20 | p-MRR | nDCG@20 | p-MRR |
| mE5-base | 0.493 | -4.2 | 0.441 | +0.3 | 0.417 | -3.5 | 0.450 | -2.5 |
| MonoT5-3B | 0.130 | -3.8 | 0.233 | +1.4 | 0.254 | +2.2 | 0.206 | -0.1 |
| FollowIR-7B | 0.163 | -0.1 | 0.404 | +6.6 | 0.379 | +7.7 | 0.315 | +4.8 |
| mT5-13B | 0.498 | +0.1 | 0.548 | +4.2 | 0.506 | +1.9 | 0.517 | +2.0 |
| RankLLaMA-7B | 0.248 | -0.9 | 0.397 | +1.3 | 0.396 | -0.1 | 0.347 | +0.1 |
| RankLLaMA-13B | 0.333 | -2.0 | 0.448 | +1.0 | 0.484 | +0.5 | 0.422 | -0.2 |
| Rank1-7B | 0.564 | +2.4 | 0.665 | +5.9 | 0.528 | +4.7 | 0.586 | +4.4 |
| Rank1-14B | 0.605 | +11.1 | 0.647 | +0.8 | 0.530 | -1.0 | 0.594 | +3.6 |
| Rank1-32B | **0.619** | **+12.1** | **0.678** | **+10.2** | **0.535** | **+8.1** | **0.610** | **+10.1** |

Table 4: Results for mFollowIR multilingual across three language subsets (Persian, Chinese, Russian). All models rerank the top 100 docs found from the mE5-base model. **We see a wide gap between RANK1 and other models, despite it not having any multilingual reranking training data.**

### 3.3 Semantic Understanding

We evaluate on NevIR which requires reasoning over negation in Table 2. Models rerank all documents, and we report scores for listwise models from van Elsen et al. (2025).

We again find that RANK1 performs extremely well, even matching GPT-4o and coming 15+ points higher than the next closest model. We see that even o3-mini in a listwise setup only performs 7 points higher.

### 3.4 Instruction-Following

We show results on the mFollowIR dataset (Weller et al., 2025a) as it illustrates both instruction-following and multilingual capabilities. Table 3 shows results on the cross-lingual setup (En-XX) and

| | Model | Score (%) |
|---|---|---|
| Listwise | RankGPT 4o-mini | 64.1 |
| | RankGPT 4o | 70.1 |
| | RankGPT o3-mini | **77.3** |
| Pointwise | RankLlama 7B | 31.6 |
| | RankLlama 13B | 43.2 |
| | MonoT5 base | 34.9 |
| | MonoT5 3B | 50.6 |
| | Rank1-7B | 65.1 |
| | Rank1-14B | 67.5 |
| | Rank1-32B | **70.1** |

Table 2: Pairwise acc. on NevIR

Table 4 on the XX-XX task. We have all models rerank the top 100 scores of a strong but small base model mE5-base. We find that RANK1 has much higher nDCG@20 scores (from 0.586 to 0.611 on the multilingual average) compared to the next best model (mT5-13B trained on Multilingual MS MARCO) with 0.517. Furthermore, when considering just instruction following metrics there is a wide gap, especially in the multilingual setting (+10.1 vs +4.8 p-MRR on the custom instruction-trained FollowIR-7B). Other models are closer on the cross-lingual version, but there still remains a notable gap between the best RANK1 and the closest other model (+7.3 vs +3.7 p-MRR).

We again find this notable considering that mT5-13B is a similar size and was trained on multilingual data. RANK1 significantly outperforms it solely with English reasoning data.

### 3.5 "Traditional" Benchmarks

**DL19**   We also evaluate DL19 scores in Table 5. All models rerank the top 100 passages found using RepLLaMA. We found that our models performed significantly worse on the initial relevance judgments (i.e. *qrels*) and our analysis quickly showed this was

| Model | Original | | Fixed |
| | Judged@10 | nDCG@10 | nDCG@10 |
| --- | --- | --- | --- |
| RankLlama 7B | 96.1 | 76.2 | 76.9 |
| RankLlama 13B | 96.1 | 77.2 | 76.9 |
| MonoT5 3B | 91.2 | 72.0 | 74.8 |
| Rank1-7B | 83.5 | 66.1 | 78.6 |
| Rank1-14B | 82.3 | 64.8 | 77.6 |
| Rank1-32B | 81.9 | 66.0 | **80.1** |

Table 5: Results on TREC DL19

due to the number of unjudged documents that our models ranks higher. As seen in Table 5, the top 10 lists of existing pointwise rerankers is almost entirely judged passages (i.e. 96.1%), whereas RANK1 finds 10-15% less (down to 81.9% passages that are judged by humans).

To remedy this, we manually annotated all top 10 documents that each model (baseline or RANK1) got wrong or were unjudged. We show examples of incorrectly or not labeled instances in Appendix D. We found that unjudged documents were mostly relevant, e.g. 70.6% of the unjudged documents for RankLLaMA 14B were relevant and 86.8% were relevant for Rank1-14b, etc.[9]  After this annotation fix[10] RANK1 models are no longer penalized for finding new documents and that performance is generally better than all other models (e.g. 78.6 vs 76.8 nDCG for RANK1-7b vs RankLLaMA 7B).

Thus, it seems that the original DL19 benchmark is no longer suitable for discriminating between the top performing approaches. Although this was examined for some TREC collections in 2022 by Voorhees et al. (2022), the largest model used in their experiments was BERT. Thus, we would encourage the community to re-evaluate old TREC collections.

**BEIR**   We show results on BEIR as a comparison in Table 6, using the datasets with less than 2k queries. All models rerank the top 100 documents found using BM25s (Lù, 2024). We find comparable but worse performance with RANK1. Although out of scope for this work, we found a large number of similar issues in BEIR datasets as we do in DL19. We discuss datasets individually in Appendix G. It appears that traditional reranking benchmarks like DL19 and BEIR – although extremely successful at driving the field forward and still useful for weaker models – are no longer as useful for distinguishing between the best rerankers.

### 3.6 Test-Time Scaling

Given the success of reasoning language models in *scaling* test time compute (i.e. getting better results with more tokens used), we also attempted several ways of using extra test-time compute to improve performance. However, using the simple budget-forcing method from s1 (Muennighoff et al., 2025) did not improve performance and actually hurt performance in our limited initial experiments. We hypothesize this may be due to the lack of difficulty in the reranking task – after all, reasoning over BRIGHT requires significantly less reasoning than the typical AIME problems used for evaluation. Alternatively, perhaps future work with new techniques are needed to induce the desired results.

---

[9]We release our new judged qrels in the Github above to help facilitate future work.

[10]We note that ideally one would re-annotate the whole dataset with new pools from many models. This would allow for future models to be able to use the same evaluation. However, given the scale of annotations, this would be out of scope for this work. Thus, we include this experiment mainly to highlight the limitations of DL19 and acknowledge that better evaluations will be needed in future work.

| Model | ArguA | ClimF | DBP | FiQA | NFCorp | SciDoc | SciFact | Touche | TrecC | Avg. |
|---|---|---|---|---|---|---|---|---|---|---|
| BM25S | 47.2 | 18.6 | 32.0 | 25.4 | 34.3 | 16.5 | 69.1 | 34.7 | 68.8 | 38.5 |
| MonoT5-3B | 42.5 | 25.4 | 44.5 | 46.5 | 37.8 | 19.3 | 76.1 | 30.7 | 79.6 | 44.7 |
| RankLLaMA-7B | 54.4 | 23.2 | 43.7 | 42.1 | 27.0 | 16.6 | 71.1 | 41.4 | 80.2 | 44.4 |
| RankLLaMA-13B | 49.3 | 24.5 | 44.9 | 44.1 | 28.1 | 18.1 | 72.7 | 39.2 | 80.8 | 44.6 |
| Rank1-7b | 42.8 | 15.0 | 38.9 | 39.5 | 36.2 | 17.2 | 77.2 | 22.8 | 81.9 | 40.9 |
| Rank1-14b | 45.3 | 16.2 | 37.4 | 37.9 | 35.8 | 17.9 | 77.0 | 27.1 | 78.2 | 41.0 |
| Rank1-32b | 57.6 | 15.8 | 40.7 | 41.8 | 36.9 | 19.6 | 76.8 | 19.9 | 81.9 | 41.7 |

Table 6: nDCG@10 on the BEIR benchmark. Models rerank the top 100 documents from BM25S (Lù, 2024). Although not in scope for this work, we find that the way many of the datasets in BEIR were constructed limits the ability to judge between the highest performing systems, similar to DL19 (see Appendix G for commentary on individual datasets).

### 3.7 How much does the reasoning chain help?

To ablate how important the reasoning chain is for RANK1 we train an ablation on the 7B model where the reasoning chain is omitted using the same data and base model. This allows us to isolate the benefits that the reasoning chain itself provides.

We find that our non-reasoning model scores an average of 17.5 on BRIGHT compared to 27.5 with Rank1-7b. This is an improvement over RankLLaMA-7B (13.9) but not the 2x improvement seen by the models with reasoning chains. Thus, we can see that the reasoning chains are providing the expressive power that is needed for these complex reranking tasks.

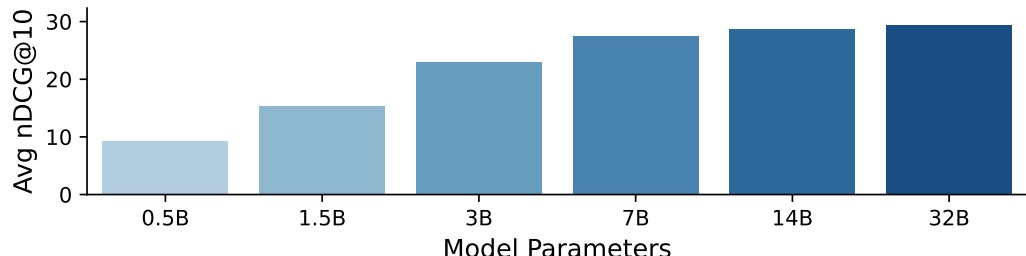

Figure 3: Average nDCG@10 across BRIGHT datasets for various sizes of RANK1. We see that with our training dataset there are diminishing returns after 7B+ parameters and that smaller models still perform very well: **Rank1-0.5b scores comparably to RankLLaMA 13B (13.9 average on BRIGHT) despite having 25x less parameters and only slightly more than BERT-sized models**.

### 3.8 Does this work at smaller sizes?

We have previously used 7B+ models, as it is the smallest size typically used for reasoning models applied to other tasks (e.g. AIME, etc.). However, for reranking, is it possible to have strong performance with smaller models?

We follow the same training recipe and train the rest of the Qwen 2.5 family of models (0.5B, 1.5B, 3B) and show the results with the main models in Figure 3. We find that larger models do better (albeit with dimishing returns), but even smaller models do very well comparatively: the 13B RankLLaMA model (13.9 average) only scores slightly higher than the 0.5B version and less than the 1.5B version.

Thus, although test-time compute is typically used with larger models, we see that – at least for reranking – even small models show large benefits.

| Model | Size (G) | SciFact | NevIR | Biology | Eng-Zho |
|---|---|---|---|---|---|
| Rank1-7B | 15.2 | 77.2 | 65.1 | 48.8 | 58.2 |
| quantized | 5.6 | 75.4 | 62.0 | 42.8 | 56.3 |
| Rank1-Llama3-8B | 16.1 | 73.2 | 61.9 | 45.8 | 58.1 |
| quantized | 5.8 | 72.9 | 57.1 | 41.5 | 54.8 |
| Rank1-14B | 29.6 | 77.0 | 67.5 | 49.3 | 61.1 |
| quantized | 10.0 | 75.6 | 66.4 | 44.6 | 58.8 |
| Rank1-Mistral-24B | 47.2 | 75.8 | 67.4 | 51.8 | 59.9 |
| quantized | 14.3 | 72.7 | 64.6 | 47.1 | 59.4 |
| Rank1-32B | 65.5 | 76.8 | 70.1 | 49.7 | 59.8 |
| quantized | 19.3 | 77.2 | 69.9 | 52.3 | 59.7 |

Table 7: Quantization results on subsets of various tasks (SciFact from BEIR, Biology subset from BRIGHT, Eng-Zho on mFollowIR-CrossLingual). We use AutoAWQ for the quantization into int4. We see that models retain most of their performance while being significantly smaller.

## 4 Model Releases

**Alternate Base Models**   Our main base models use the Qwen 2.5 series due to their strong performance and varying parameter sizes. However, to show that this approach works on other base models, we train a version with Llama 3.1 8B (AI@Meta, 2024) and Mistral Small 2501 24B.[11] Similar to the Qwen models, we start from the base versions, not the instruct versions. We show the results on selected benchmarks in Table 7 where we can see that our approach holds for other base models with performance generally increasing as they scale (i.e. 24B is better than 8B). We also see that Llama 3.1 8B slightly underperforms Qwen 7B.

**Quantization**   IR models typically require a heavier inference workload compared to many other LLM applications due to the number of passages to rerank. However, many common rerankers (MonoT5, RankLLaMA) do not take advantage of modern inference capabilities such as quantization.

We quantize each of our models using AutoAWQ (Lin et al., 2023) and compare performance before and after. We see that although performance drops slightly, the model size is significantly smaller, enabling **all models** (including the 32B version) to be run on one 24GB GPU. Despite the performance loss, quantized RANK1 models still significantly outperform the baselines on reasoning and instruction-following while being 1/3rd of the size.

## 5 Related Work

### 5.1 Advanced Reasoning and Instruction Following in IR

In the last couple years, retrieval systems have started to move beyond simple phrase-based semantic matching, to more complex information retrieval tasks. This has included a focus on new benchmarks: such as reasoning in retrieval (Su et al., 2024; Zhao et al., 2024; Xiao et al., 2024), instruction-following capabilities (Weller et al., 2024a; Oh et al., 2024), and retrieval for RAG systems (Lawrie et al., 2024; Mayfield et al., 2024).

On the modeling side, we have seen a surge of interest in models that can understand the meaning behind the user's query rather than doing phrase-based matching: this includes models like Instructor (Su et al., 2022), TART (Asai et al., 2022), GritLM (Muennighoff et al., 2024), FollowIR (Weller et al., 2024a), Gecko (Lee et al., 2024), and Promptriever (Weller et al.,

---

[11]Mistral Small can be found at `https://huggingface.co/mistralai/Mistral-Small-24B-Base-2501`

2024b). These models typically use instruction-based data in their training data, so that they learn to adapt to new user instructions. Other works have built multi-agent systems with LLMs that do reasoning (Niu et al., 2024; Ji et al., 2024). However, for RANK1 we use a single model and not provide any instruction-based training data and fine-tune from the base (non-instruct) version of the LMs – despite this our model shows SOTA ability.

## 5.2 Reasoning Language Models

Reasoning language models were introduced by OpenAI with their o1 model (Jaech et al., 2024). These models showed significantly improved performance on tasks that needed reasoning, such as math, logic, and programming. Since their release, many others have trained similar style models, including Google's Gemini Flash Thinking and Deepseek's R1. Notably, R1 is the only reasoning model that provides reasoning chains through APIs and is the only open-weights model.

Other than the impressive performance gains of these models, one additional feature is that models can quickly learn to emulate stronger models through basic supervised fine-tuning, rather than the more complex reinforcement learning pipelines that are typically used. We take advantage of this capability to train RANK1 using a simple training process.

There have also been a flurry of works in the open-source space, both before and after o1 on reasoning language models and systems, focused on reproduction (Snell et al., 2024; Muennighoff et al., 2025), calibration and confidence (Jurayj et al., 2025), agentic capabilities with explainable reasoning traces (Weir et al., 2022; 2024), and much more. We expect this line of work to continue and likely improve these models in retrieval as well.

## 6 Limitations and Future Work

**Overthinking**   Like other reasoning models, RANK1 can make mistakes and it can be surprising to see the model's reasoning chain come close to a correct answer only to change its mind. We also found that RANK1 can be particularly stringent in marking passages as true. For example, on a TREC COVID query about the origins of COVID-19 it marked every single passage as non-relevant, since none mentioned the specific wet market in Wuhan. However, when given a prompt to assume that the user had no information about COVID-19, it was able to adapt better (although not perfectly). We observed this tendency to "overthink" when using the model interactively, as it would already know the answer and was looking for a very specific phrase. We expect that this could be reduced with data that specifically trains the model to calibrate this.

**Inference Speed**   As a reasoning model using test-time compute RANK1 is slower than a model with only a classification head (i.e. RankLLaMA). In practice this can be somewhat mitigated by the usage of modern paged attention libraries like vLLM (Kwon et al., 2023) which we use, and also quantization techniques. Nonetheless, there is no getting around the fact that using test-time compute requires spending more compute than non-test-time compute models. Despite this additional compute usage, we see that users are willing to wait longer for quality search results, as illustrated by the popularity of the Deep Research products from Google and OpenAI.[12]

**Future Work**   RANK1 also brings many more exciting areas for future work (and concurrent work (Yang et al., 2025)). As highlighted in the experiments section, it does extremely well despite the relatively small and non-diverse training data. Some of the promising areas of future work are:

- Fine-tuning with RL: although supervised fine-tuning works, it does not optimize for the final answer. It is likely that RL-based approaches would be able to add additional rewards/penalties that could better align the final prediction to the label.

---

[12]https://openai.com/index/introducing-deep-research/ and https://blog.google/products/gemini/google-gemini-deep-research/

- Listwise reasoning rerankers: although pointwise models are more efficient and more parallelizable, listwise rerankers are generally more performant since they can see many documents at once. Simply gathering new data should enable this to be very successful.
- Multilingual and instruction-tuned versions: RANK1 was solely trained on English and non-instruct data and still showed strong results in these settings. Training on a more curated set of datasets will likely significantly improve the performance on these tasks.

We expect to see many future applications of test-time compute applied to retrieval with great success, as the recipe is both simple and effective.

## 7 Conclusion

We build the first reasoning reranker model that uses test-time compute, RANK1. We do so by collecting 600k+ examples from the reasoning language model R1, fine-tuning on its reasoning traces. Despite only using English MS MARCO data and training from base (non-instruct-tuned) language models, RANK1 shows state of the art reasoning and instruction following capabilities, even in multilingual settings. Overall, RANK1 introduces a new category of reranking models that enable a wide variety of more complex IR tasks.

## 8 Ethics Statement

RANK1 is a more generative form of many rerankers currently used in information retrieval. As such, it introduces the possibility of generating flawed text (just like LMs). This includes incorrect and hallucinated information, the potential for toxic language, biased output, and other such flaws. These same flaws may also be present in the R1-generated dataset used to train these models.

As an improved reranker, is also increases the risk when used for illicit activities. As shown by (BehnamGhader et al., 2025) models that are better at ranking and following instructions can be used more effectively to retrieve information that leads to illicit purposes. However, most of the use cases of information retrieval are positive; such is the balance when using a dual-use technology. We would encourage users to use improved retrieval capabilities for positive purposes.

## Acknowledgments

This work has been supported by both DARPA SciFy and the U.S. National Science Foundation under grant 2204926. Any opinions, findings, and conclusions or recommendations expressed in this article are those of the authors and do not necessarily reflect the views of the National Science Foundation or DARPA. OW is supported by an NSF GRFP fellowship.

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

---

**Prompt for Inference**

Determine if the following passage is relevant to the query. Answer only with 'true' or 'false'.

Query: query

Passage: document

<think>

---

Figure 4: Prompt used to generate data with R1 and also for inference with RANK1.

## A  Training and Hyperparameter Details

We use nodes of 4x80GB H100 machines for training. Models are trained for up to 2 epochs or until 3 days of training (for the 32B model).

We fine-tune with LLaMA-Factory, using LoRA on all parameters with rank 32 and alpha 64. We use a learning rate of 1e-4 and an effective batch size of 128. We use early stopping based on the Bright Biology and NevIR scores. Models use prompts for BEIR and non-stackexchange cases so that they can understand the task (Appendix E).

Inference is done on 1 H100 80GB GPU. Baselines use default hyperparameters for max length and fp16 (and there are no other parameters).

## B  Data Analysis from mT5 Negatives

We sample 20 of the instances from mT5 that were "hard negatives" and manually evaluated them. We found that 12 of them were in fact positives and thus false negatives. This helps to explain why filtering these out made such a large difference in the training data, as roughly 2/3rds of these mT5 negatives were actually positive, making the training process more noisy.

## C  Prompt for R1

We use the following prompt (Figure 4) to generate data for R1 (and use the same template for RANK1).

## D  Examples of Unjudged and Incorrect DL19 Labels

We re-annotated the top 10 passages for each model that got an incorrect or unjudged label. This was 295 labels. We found that none of the labels changed from correct to incorrect, but some labels went from incorrect to correct (Table 8).

We also show some examples of these cases in Table 9. The incorrect labels affected all models roughly equally, as seen previous in Table 8 at around 6% of the top 10 documents, while the unjudged documents were mostly found by RANK1.

## E  Prompts for specific datasets

Table 10 shows the data-specific prompts used for BEIR and the non-stackoverflow BRIGHT subsets.

| Model | Unjudged→Relevant | Changed→Correct |
|---|---|---|
| MonoT5-3b | 50.00% (19/38) | 6.74% (29/430) |
| RankLLaMA-13B | 70.59% (12/17) | 6.28% (27/430) |
| RankLLAMA-7B | 70.59% (12/17) | 6.74% (29/430) |
| Rank1-7B | 85.92% (61/71) | 5.35% (23/430) |
| Rank1-14B | 86.84% (66/76) | 6.05% (26/430) |
| Rank1-32B | 88.46% (69/78) | 6.51% (28/430) |

Table 8: Qrel changes in the DL19 annotations

---

**Example 1**

**Query:** what is physical description of spruce
**Passage:** Spruces are large trees, from about 20–60 metres (about 60–200 feet) tall when mature, and can be distinguished by their whorled branches and conical form. The needles, or leaves, of spruce trees are attached singly to the branches in a spiral fashion, each needle on a small peg-like structure.
**Original label:** 0 (non-relevant)
**New label:** 3 (relevant)

**Example 2**

**Query:** causes of left ventricular hypertrophy
**Passage:** High blood pressure may also bring on heart failure by causing left ventricular hypertrophy, a thickening of the heart muscle that results in less effective muscle relaxation between heart beats. This makes it difficult for the heart to fill with enough blood to supply the body's organs, especially during exercise, leading your body to hold onto fluids and your heart rate to increase.
**Original label:** unjudged
**New label:** 3 (relevant)

**Example 3**

**Query:** what are the social determinants of health
**Passage:** © Zoltan Balogh. The social determinants of health (SDH) are the conditions in which people are born, grow, work, live, and age, and the wider set of forces and systems shaping the conditions of daily life.
**Original label:** 1 (on topic)
**New label:** 3 (relevant)

**Example 4**

**Query:** example of monotonic function
**Passage:** Overview of the exponential function. The exponential function is one of the most important functions in mathematics (though it would have to admit that the linear function ranks even higher in importance). To form an exponential function, we let the independent variable be the exponent. A simple example is the function $f(x) = 2^x$. ...
**Original label:** unjudged
**New label:** 3 (relevant)

Table 9: Examples of incorrect labels and unjudged documents in the DL19 annotations.

| Dataset | Prompt |
|---|---|
| `SciFact` | Claim: FILL_QUERY_HERE<newline><newline>A relevant passage would provide evidence that either **supports** or **refutes** this claim. A passage with any information on any related subpart should be relevant. |
| `ClimateFEVER` | Claim: FILL_QUERY_HERE<newline><newline>A relevant passage would provide evidence that either **supports** or **refutes** this claim. A passage with any information on any related subpart should be relevant. |
| `TRECCOVID` | FILL_QUERY_HERE If the article answers any part of the question it is relevant. |
| `ArguAna` | I am looking to write an essay and need to find counterarguments against this statement:<newline><newline>FILL_QUERY_HERE<newline><newline>Does this passage have any counterargument or evidence that could be used to help me? |
| `DBPedia` | I am looking to write an essay on this topic and need as much related background information to help me. The topic is:<newline><newline>FILL_QUERY_HERE<newline><newline>If the passage provides any background information that could be connected it is relevant. |
| `FiQA2018` | FILL_QUERY_HERE Find a passage that would be a good answer from StackExchange. |
| `NFCorpus` | Topic: FILL_QUERY_HERE<newline><newline>Given the above topic, I need to learn about all aspects of it. It does not need to be directly relevant, only tangentially informational. Please mark as relevant any passages with even weak connections. I need to learn fast for my job, which means I need to understand each part individually.<newline><newline>Again remember, any connection means relevant even if indirect. So if it is not addressed, that is okay – it does not need to be explicitly.<newline><newline>Find me passages with any type of connection, including weak connections!!!! |
| `Touche2020` | FILL_QUERY_HERE **any** arguments for or against |
| `SCIDOCS` | papers that could be cited in FILL_QUERY_HERE. Anything with even indirect relevance should be relevant. This includes papers in the same broader field of science |
| `BrightRetrieval aops` | Find different but similar math problems to FILL_QUERY_HERE<newline><newline>A document is relevant if it uses the same class of functions and shares **any** overlapping techniques. |
| `BrightRetrieval theoremqa questions` | Find a passage which uses the same mathematical process as this one: FILL_QUERY_HERE |
| `BrightRetrieval leetcode` | I am looking to find different problems that share similar data structures (of any kind) or algorithms (e.g. DFS, DP, sorting, traversals, etc.). I am looking for problems that share one or both of these similarities to this:<newline><newline>FILL_QUERY_HERE<newline><newline>Does this passage share any similarities? e.g. if there was a textbook on leetcode problems, this would be in the same book even though it could be in a different chapter. |
| `BrightRetrieval pony` | I will use the programming language pony. Problem: FILL_QUERY_HERE<newline><newline>But to solve the problem above, I need to know things about pony. A passage is relevant if it contains docs that match **any** part (even basic parts) of the code I will have to write for the above program. |
| `BrightRetrieval` | Can you find background information about the concepts used to answer the question:<newline><newline>FILL_QUERY_HERE<newline><newline>A passage is relevant if it contains background information about a **sub-concept** that someone might cite/link to when answering the above question. |
| `BrightRetrieval theoremqa theorems` | Find a passage which uses the same mathematical process as this one: FILL_QUERY_HERE |

Table 10: Dataset-specific prompts used in the BEIR and non-StackExchange subsets of BRIGHT.

## F  Things we tried that didn't work

We tried a few other things that didn't work:

- We tried to calibrate the scores better by adding a ModernBERT model (Warner et al., 2024; Weller et al., 2025b) on top of the outputs of the reasoning chain. However, this ended up performing worse. It is possible that with better curation this would be an effective approach however.

- We tried adding extra loss to the last token of the next token prediction loss in LLaMA-Factory. However, this resulted in sub-par performance, likely because predicting the last true/false token at the end of reasoning chain is fairly easy (as the model already states beforehand "the answer is X" leaving little room for doubt).

## G  Noise in BEIR benchmarks

We discuss here a few of the issues with individual BEIR datasets which cause them to be noisy and less accurate at judging between highly effective systems.

**SciFact**  In the original SciFact work (Wadden et al., 2020) the authors have three labels: support, refute, or not enough information (NEI). However, when incorporated into BEIR, the NEI queries are still included. As there was not evidence to support these queries (as determined by the original authors) these queries are effectively noise. In practice, what it means is that tests model's ability to find the top ranked BM25 document, which is guaranteed to not have enough information to either refute or support the claim. The large number of these queries add a significant amount of noise.

**FiQA2018**  FiQA is scraped from Financial Stackexchange (Maia et al., 2018). However, during the original scrape the creators did not collect the post's details. Thus, the retrieval setup is to take the posts title and search for the top answer. Yet, in many cases the user clarified important details in the post that entirely changed the meaning of the query. Without this additional information for some queries it is impossible for someone to determine what the best answer is as the post details asked many other questions that were different from the title of the post.

**DBPedia**  DBPedia has many partially relevant labels, that give credit for finding non-relevant information. For example, the entire passage with a relevance of "1" for the query "Eiffel Tower" is "The year 1989 in architecture involved some significant architectural events and new buildings." However, nothing about the Eiffel Tower is connected to 1989. After digging, the only connection is its the 100 year anniversary of it being built – but the passage does not mention this. There are many such examples in the dataset (and more documents judged "1" than "2"), contributing to the noise.

**Touche2020**  Touche2020 has been well examined by Thakur et al. (2024). They created a much cleaner version of the data, however, it is not the "standard" evaluation set in BEIR.

**Other datasets with partial relevance**  Many of the other datasets give credit for partial relevance in a similar manner to DBPedia. This includes TREC COVID (which has so many real positives that it is a non-issue for the top-10), NFCorpus, and SciDocs. For many of these datasets, it is very difficult for even a human to match relevance: e.g. on NFCorpus you would have to guess any potential link that had been on that website and was even ancillary related to the title. An example is the query "How Fruits and Vegetables Can Treat Asthma" which matches a document with the title "Effect of a single high-fat meal on endothelial function in healthy subjects." and does not discuss asthma at all. The reranker would have to assume that any passage discussing food or asthma separately in any context would be relevant. Although one intuition is that these documents should rank higher than completely non-relevant documents, for RANK1 it treats them as the same (as they are both

equally non-relevant to the query). This may be suboptimal for some approaches, but for today's RAG use cases returning only actual relevant documents seems more useful.

