# OpenReview forum: "Rank1: Test-Time Compute for Reranking in Information Retrieval"
_colmweb.org/COLM/2025/Conference — COLM 2025_

### Official Review · Reviewer_Lnne · 2025-05-02

**Rating:** 7
**Confidence:** 4
**Ethics Flag:** 1

**Summary:**

This work presents Rank1, a passage reranker that leverages inference-time compute and reasoning traces. The method uses the MS MARCO dataset as the training set and collect reasoning traces from a powerful reasoning LM (DeepSeek-R1). The paper considers different sources of hard negatives and found certain models work better than others. Then, the collected training dataset is distilled into smaller models, such as the Qwen-2.5 models. Finally, the experiments include comprehensive datasets ranging from traditional benchmarks such as BEIR to more recent reasoning-heavy tasks like BRIGHT. The experimental results are strong and the baselines are comprehensive.

**Questions To Authors:**

- Sec 3.6 mentions that “using the simple budget-forcing method” does not help, where are the results for this located?
- What are the effects of quantization on the inference speed (Table 7)?

**Reasons To Accept:**

- The paper is well-written and well-motivated as passage reranking is a popular and critical component of modern retrieval systems. Furthermore, the strong results are demonstrated on tasks on diverse tasks and benchmarks from semantic search to reasoning-heavy and multilingual tasks.
- The experimental details are clear and convincing; the baselines are consisted of relevant recent models.
- The produced artifacts, the Rank1 models, may be useful for the community in both future research and practical applications.
- The paper also discusses the noise in traditional benchmarks that may be insightful to the community.

**Reasons To Reject:**

- The paper is not particularly novel in terms of its technical contributions—there are many recent works that collect and distill reasoning traces into smaller models. This may not be a big issue as it’s applied in a different setting (reranking).
- Although the paper includes many details and takeaways from preliminary experiments, it can benefit from additional explanations (perhaps in the appendix if there are not space in the main text for the curious readers). For example, what percentage of hard negatives from mT5 are actually false negatives, some human analysis may give more insight to the training data.

---

> ### Author Response · Authors · 2025-05-31
>
> Thank you for recognizing that our work is “well-written and well-motivated” and that the “experimental details are clear and convincing”!
>
> > The paper is not particularly novel in terms of its technical contributions—there are many recent works that collect and distill reasoning traces into smaller models. This may not be a big issue as it’s applied in a different setting (reranking).
>
> We agree that our main novelty is in the information retrieval setting
>
> > Although the paper includes many details and takeaways from preliminary experiments, it can benefit from additional explanations (perhaps in the appendix .. For example, what percentage of hard negatives from mT5 are actually false negatives
>
> Thank you for the suggestion! As mT5 mined hard-negatives were done automatically we didn't have any human annotations. However, upon your suggestion we sampled 20 of them and 12 were false negatives.
>
> We will add this to our paper and also be sure to add more explanations throughout. This helps to explain why filtering these out made such a large difference in the training data, as roughly 2/3rds were actually positive.
>
> > What are the effects of quantization on the inference speed (Table 7)?
>
> We have an issue filed with the quantization repo as the speed differences are negligible between the quantized and non-quantized. We think this may be fixed in the future, hence why we left out those specific numbers and only reported size decreases. We hope that it will be faster after the issue is resolved – but this issue is not central to the paper’s claims.
>
> > Sec 3.6 mentions that “using the simple budget-forcing method” does not help, where are the results for this located?
>
> We do not include any specific results in the paper. We tried two BRIGHT subsets (Leetcode and Biology) with various budget forcing methods and could not find any simple strategy to boost performance – and the inference was quite expensive with multiple budget forcing rounds. Thus, we did not pursue this in depth and don’t have any official set of numbers to report.
>
> We will update this section in the paper to be more clear that this was a very limited set of initial experiments.

---

> > ### Comment · Reviewer_Lnne · 2025-05-31
> >
> > Thanks for the response. Excited to see all the things the authors mentioned in the final draft, and I don't have any major concerns. I will maintain my positive assessment.

---

### Official Review · Reviewer_iKPU · 2025-05-11

**Rating:** 7
**Confidence:** 3
**Ethics Flag:** 1

**Summary:**

The paper focuses on reranking the retrieved passages based on their relevance to the question. To address this task, the authors prompt LLM to reason and verify whether a passage is relevant to the question, and construct a new dataset whose sample contains reasoning text and true/false answers. Models trained on the proposed dataset outperform previous ranking methods on recent and traditional benchmarks.

**Questions To Authors:**

The authors can provide some output examples to present the reasoning process of the trained models.

**Reasons To Accept:**

The paper provides a new dataset for reasoning on the relevance. The authors carefully generate and filter the data, enabling them to help further work in this field.

**Reasons To Reject:**

- The compared method is too weak to validate the effectiveness. RankLLaMA is based on LLaMA-2, which is much weaker than Qwen2.5. The authors should compare with stronger models or use recent general LLMs as baselines.

- The experiments neglect how the retrieval improves the final answering accuracy. The origin BRIGHT benchmarks evaluate Claude's answering accuracy with and without the reranked passage.

---

> ### Author Response · Authors · 2025-05-31
>
> Thank you for recognizing that we create a new dataset that will “help further work in this field”!
>
> > The compared method is too weak to validate the effectiveness. RankLLaMA is based on LLaMA-2, which is much weaker than Qwen2.5. The authors should compare with stronger models or use recent general LLMs as baselines.
>
> **We do compare with a Qwen 2.5 baseline without the reasoning traces in Section 3.7.** We see that our model using the reasoning traces beats it by +10 points!
>
> As we were the first to use reasoning models for information retrieval there are no other reasoning baselines that are stronger. And, at the time RankLLaMA was the best non-reasoning reranker. If you are aware of any other non-reasoning rerankers we should compare with, please let us know! However, we have tried to be as thorough as we could.
>
> > The experiments neglect how the retrieval improves the final answering accuracy. The origin BRIGHT benchmarks evaluate Claude's answering accuracy with and without the reranked passage.
>
> We agree that information retrieval models are often used for RAG, however, that is out of scope for our work  which is only focused on information retrieval alone.
>
> Information retrieval - without RAG - is still one of the most used global technologies with over 16 billion searches a day just on the Google search engine alone, let alone other web search engines or search usage in other applications such as email, etc.
>
> Our work focuses on improving this key technology through the use of reasoning language models.

---

> > ### Comment · Reviewer_iKPU · 2025-06-05
> >
> > Thanks for the response. It has addressed my concerns about the baseline results. I have raised my rating and advise adding the Qwen2.5 result to Table 1.

---

### Official Review · Reviewer_Znd7 · 2025-05-12

**Rating:** 7
**Confidence:** 3
**Ethics Flag:** 1

**Summary:**

The paper introduces Rank1, a model that leverages test-time compute for reranking in information retrieval (IR). Rank 1 distills reasoning traces from DeepSeek R1; in particular, the authors prompt DeepSeek R1 using the MS MARCO collection, gathering ~600k query-reasoning traces, which they use to finetune Qwen 2.5 using LoRA. Through a thorough experimental evaluation across scales, the authors present strong performance gains across a variety of benchmarks and baselines, gains that are maintained after quantization.

**Reasons To Accept:**

- The paper tackles an interesting problem, demonstrating that it is possible to distill reasoning chains into ranking models, resulting in efficient IR models, that also provide reasoning.

- The paper has very strong empirical results; the authors present results across a wide variety of benchmarks, showcasing the advantages of Rank1. They have also included results across different scales (from 0.5B to 32B), observing that their method achieves better nDCG@10 performance across the BRIGHT dataset, with diminishing returns after 7B parameters. They also explore quantization, finding that it has minimal effect across many tasks.

- The authors also identified multiple mislabeled or entirely undjudged relevant documents in a subset of DL19 and BEIR, arguing that this may be indicative of why such traditional reranking benchmarks are cannot distinguish between current SOTA rerankers.

**Reasons To Reject:**

- When evaluating on “traditional” benchmarks, the authors manually relabel the retrieved passages that were either wrongly labeled or previously unjudged. I believe this still inserts bias. Most unjudged or mislabeled hits are in Rank 1’s top-10, while the baselines’ top-10 lists were already almost fully judged. Therefore, the relabeling inflates Rank 1’s scores far more than theirs and makes the gain look larger than it really is.

- Although the current approach (supervised finetuning) yields significant gains and is simple, I think that, as the authors also point out in the future work section, exploring a RL based approach would -perhaps- result in significantly more important improvements.

- The paper assumes familiarity with IR jargon—terms like “qrels,” “DL19,” and “point-wise vs. pair-wise” appear without definition. I think that adding a quick one-line definition the first time each term appears (or a small abbreviation table) would help readers outside the IR sub-community.

---

> ### Author Response · Authors · 2025-05-31
>
> Thank you for your review and for noting that our work “tackles an interesting problem”, has “strong empirical results … across a wide variety of benchmarks”!
>
> > When evaluating on “traditional” benchmarks, the authors manually relabel …I believe this still inserts bias. Most unjudged or mislabeled hits are in Rank 1’s top-10 … Therefore, the relabeling inflates Rank 1’s scores far more than theirs and makes the gain look larger than it really is.
>
> We agree that Rank1 had the most unjudged documents in the top 10 and thus it had the most to gain from annotating unlabeled documents. However, the alternative is penalizing models like Rank1 that found documents that older search systems didn’t find (and thus wasn’t labeled). When those unjudged documents really are relevant, it also seems unfair to penalize it. Either way there are problems with using the dataset for evaluation – to truly be fixed, the whole dataset would need to be re-annotated.
>
> Thus, we treat DL19 mainly as an experiment to show that it (and similar old datasets) are not helpful for evaluation of strong models. We will make this more clear and be sure to highlight this tradeoff regarding re-annotation in the paper, thank you for the suggestion!
>
> > The paper assumes familiarity with IR jargon—terms like “qrels,” “DL19,” and “point-wise vs. pair-wise” appear without definition. I think that adding a quick one-line definition the first time each term appears (or a small abbreviation table) would help readers outside the IR sub-community.
>
> This is a great suggestion - thank you for the feedback! We will be sure to update the paper with this.

---

> > ### Comment · Reviewer_Znd7 · 2025-05-31
> >
> > Thank you for your response. I agree that to fully mitigate the dataset-related issues, the latter should be re-annotated, fixing all the mislabeled samples. Thus, having no additional concerns, I will maintain my positive score.

---

### Decision · Program_Chairs · 2025-07-08

**Decision:**

Accept

**Comment:**

This paper leverages test-time compute for a reranking model (called Rank1) by distilling reasoning traces from DeepSeek R1 into smaller models. The reviewers gave consistent positive reviews. The work demonstrates clear technical merit with comprehensive evaluation across traditional and reasoning-heavy benchmarks, showing substantial improvements over existing rerankers, and contributes a valuable 600k query-reasoning trace dataset to the community. While concerns exist about potential bias from manual relabeling of unjudged documents and limited technical novelty beyond applying established distillation methods to retrieval, the fundamental contribution of achieving state-of-the-art performance with explainable reasoning represents meaningful progress for the retrieval community. As some reviewers pointed out, it would be better to include some details and takeaways from preliminary experiments in the final version.